# Mechanical Properties, Crack Width, and Propagation of Waste Ceramic Concrete Subjected to Elevated Temperatures: A Comprehensive Study

**DOI:** 10.3390/ma15072371

**Published:** 2022-03-23

**Authors:** Hadee Mohammed Najm, Ominda Nanayakkara, Mahmood Ahmad, Mohanad Muayad Sabri Sabri

**Affiliations:** 1Department of Civil Engineering, Zakir Husain Engineering College, Aligarh Muslim University, Aligarh 202002, India; 2Department of Civil Engineering, Xi’an Jiaotong-Liverpool University, Suzhou 215000, China; ominda.nanayakkara@xjtlu.edu.cn; 3Department of Civil Engineering, Faculty of Engineering, International Islamic University Malaysia, Jalan Gombak, Selangor 50728, Malaysia; ahmadm@iium.edu.my; 4Department of Civil Engineering, University of Engineering and Technology Peshawar (Bannu Campus), Bannu 28100, Pakistan; 5Peter the Great St. Petersburg Polytechnic University, 195251 St. Petersburg, Russia; mohanad.m.sabri@gmail.com

**Keywords:** Waste Ceramic Concrete, Mechanical Performance, Fire Resisting, ANN, MLR

## Abstract

Waste ceramic concrete (WOC) made from waste ceramic floor tiles has several economic and environmental benefits. Fire is one of the most common disasters in buildings, and WOC is a brittle construction material; therefore, the mechanical properties of WOC structures under high temperatures should be considered. According to previous studies, hybrid fiber can further reduce damage to concrete under high temperatures. Meanwhile, crack width and propagation are among the key characteristics of concrete materials that need to be considered, but few studies have focused on their behavior when subjected to elevated temperatures. The new concrete materials proposed by the authors are WOC and WOC-Hybrid. WOC was prepared with Natural Coarse Aggregates (NCA), Natural Fine Aggregate (NFA), Ordinary Portland Cement (OPC 43 grade), and ceramic waste tiles with 20% replacements for coarse aggregates, 10% replacements for fine aggregates, and 10% replacement for cement. In contrast, WOC-Hybrid was prepared with the addition of hybrid fiber (1% crimped steel fiber and 1% polyvinyl alcohol fiber) in WOC. The specimens were exposed to temperatures of 100–300 °C, and then the specimens were tested for tensile and compressive strength. The present study aims to find a new method to improve concrete resistance to elevated temperatures at the lowest costs by experimental and computational analysis via machine learning models. The application of machine learning models such as artificial neural networks (ANN) and multiple linear regression (MLR) was employed in this study to predict the compressive and tensile strength of concrete. The linear coefficient correlation (R^2^) and mean square error (MSE) were evaluated to investigate the performance of the models. Based on the experimental analysis, the results show that the effect of hybrid fiber on the crack width and propagation is greater than that on the crack width and propagation of WOC and PC after exposure to high temperatures. However, the enhanced effect of hybrid fiber on the mechanical properties, rack width, and propagation decreases after subjecting it to a high-temperature treatment, owing to the melting and ignition of hybrid fibers at high temperatures. Regarding the computational analysis, it was found that the developed MLR model shows higher efficiency than ANN in predicting the compressive and tensile strength of PC, WOC, and WOC-Hybrid concrete.

## 1. Introduction

Buildings constitute the majority of built infrastructure and play a pivotal role in the socio-economic development of a country. Most buildings are designed to last for several decades and provide residential and functional operations to a large number of inhabitants throughout their design life. During this long-time span, buildings are subjected to several natural and manmade hazards which can cause a partial or complete collapse of the building and incapacitation of building operations. Such destruction or incapacitation in the event of a hazard can jeopardize the lives and safety of inhabitants and can cause significant direct and indirect monetary losses. Hence, buildings are designed to withstand actions from numerous anticipated hazards to ensure life and structural safety during their design life, and fire represents one such extreme hazard that can occur in buildings.

Therefore, the resistance of concrete structures to rising temperatures is a major global concern today. It requires a strong and unified effort to arrive at a heat-resistant economic concrete using readily available resources. A promising source of low-cost materials is the growing amount of waste generated as a result of excessive urbanization, global economic development, and rapid industrialization. Casares, M.L. et al. [1] noted that “waste materials contain toxic chemicals that are harmful to the environment and humans and traditional waste management practices commonly employed generally increase environmental contamination”. Therefore, the aim is to find solutions to the increasing quantity of waste dumped into the natural environment by finding the optimum use of waste materials. One such process uses concrete materials with particular characteristics, reducing prices and improving their characteristics [2].

Extreme heat causes dramatic physical and chemical changes in concrete, resulting in its deterioration [3]. Although concrete is more highly thermally resistant than other construction materials, critical deterioration occurs when exposed to high temperatures, such as in a fire. When concrete is heated to a high temperature, several irreversible physical and chemical changes occur. Industrial by-products and solid wastes, such as waste ceramics, could be used in concrete as a substitute material, reducing the concrete industry’s negative environmental impact. As a result, it is necessary to investigate the strength characteristics of concrete containing waste ceramic.

When conventional concrete is exposed to heat, exceptionally elevated temperatures cause it to lose significant strength or even wholly degrade. The presence of limestone rocks and ordinary cement in concrete is the main reason for the cracks and the release of certain gases when exposed to elevated temperatures. However, limestone is the major reason that contributes to this process. Ordinary rocks are primarily composed of lime. Normal lime (CaCO_3_) degrades to form quicklime (CaO), which is extremely hydrophilic. When lime comes into contact with water or air humidity, it will absorb a large amount of water. Subsequently, its volume will be increased. This increase in volume results in cracks in the rocks, thus causing the concrete to collapse [4]. As an alternative to such heavy reliance on limestone concrete mixes, previous research has used various waste materials, such as ceramic wastes, to increase the material’s heat resistance. Ceramic is a material that resists pressure, humidity, and high temperatures, so it is used in wall and floor façades; as a result, it is also effective at improving the properties of concrete [5,6,7].

The previous study conducted by the same authors proved concrete mixes containing ceramic waste materials such as cement and aggregate replacement had been shown to exhibit enhanced tensile and flexural strength and torsion [8]. Moreover, Kulovaná, T. et al. [9] proved that “ceramics ameliorate concrete resistance to chemical and heat attacks while decreasing concrete water absorption”. Thus, in general, studies indicate that using ceramic waste in concrete is feasible, though there may be some differences in the concrete’s properties. Additionally, ceramics increase the resistance of concrete to chemical and thermal attacks while decreasing its water absorption [10].

Recent research has focused on using ceramic wastes of various sizes as a full or partial replacement for concrete aggregates, to reduce the negative environmental effects of waste ceramics [11]. In addition, other studies have shown that powdered ceramic waste can be used in place of Portland cement as a pozzolanic material [9]. In general, studies indicate that using ceramic waste in concrete is feasible, though there may be some differences in the concrete’s properties.

One reason for concrete damage and cracking at elevated temperatures is thermal stress concentration due to heterogeneous heat distribution in concrete. Researchers used metallic and non-metallic fiber in concrete exposed to high temperatures to counteract this effect; for example, Gao, D. et al. [12] studied the effect of PP fibers in dosages 0.5, 1, 1.5, and 2% (by volume) on tensile strength behavior under elevated temperatures of 100 °C, 450 °C, and 650 °C. Gao, C. et al. [13] studied the effect of PP fibers in dosage 0.5% (by volume) on tensile and compressive fibers’ reinforced concrete under elevated temperatures of 200 °C and 400 °C. Adding steel fibers to concrete improved its physical and chemical properties when exposed to elevated temperatures. Poon, C.S. et al. [14] observed significantly improved concrete compressive strength when steel fibers were used in high-performance concrete. Aslani, F., and Samali, B. [15] studied the effect of steel fiber in dosages 0.5, 1, 1.5, and 2% (by volume) on compressive and tensile strength of high-performance concrete at high temperatures of 300 °C, 500 °C, and 800 °C. Aydn, S., et al. [16] studied the behavior of normal and high-performance concrete reinforced with PP and steel fibers tested under elevated temperatures of 200 °C, 400 °C, and 600 °C.

Non-metallic fiber has been successfully used to increase the ductility of concrete on a large scale. Due to the low Young’s modulus of non-metallic fibers such as polyvinyl alcohol fiber, they cannot prevent the formation and propagation of cracks under high-stress conditions, but they can bridge large cracks [17,18]. Recently, it was discovered that a variety of fibers can also be used to enhance the residual properties of concrete following exposure to elevated temperatures. Numerous studies conducted by various authors [19,20] demonstrate that incorporating non-metallic fibers into concrete improves its thermal stability. The use of polyvinyl alcohol (PVA) fibers has been shown to significantly reduce spalling and cracking and to increase residual strength [21,22]. Since the fibers melt at approximately 160–170 degrees Celsius, they form expansion channels. Noumowe, A. [23] concluded that “additional porosity and small channels created by non-metallic fibers melting may reduce the internal vapor pressure of the concrete, thereby decreasing the likelihood of spalling”. However, we observed that non-metallic fibers had a negligible or even detrimental effect on the residual performance of heated concrete. The increased porosity caused by the melting of PVA fibers may reduce the residual mechanical properties of concrete.

Therefore, in the present study, the authors propose two new types of concrete: WOC and WOC-Hybrid. Many considerations have been made based on the new generation concrete, as follows: (1) PVA + crimped steel fiber is used to improve the performances of concrete both before exposure (compressive, tensile, flexural, and torsion strength) and after exposure (compressive and tensile strength) to elevated temperatures. (2) The inclusion of ceramic waste material is mainly for environmental protection and ceramic tiles have lower thermal conductivity and better fire resistance after exposure to high temperatures. (3) A beneficial interaction exists between hybrid fiber (PVA + CR) and ceramic, as mentioned above. Improved resistance to elevated temperatures makes WOC-Hybrid concrete useful in fire-resistant structures.

The authors have conducted several tests to investigate the behavior of the proposed WOC and WOC-Hybrid. This paper presents a study on the effect of ceramic material (ceramic powder, fine and coarse ceramic aggregate) and hybrid fiber (PVA and CR fibers) on the mechanical behaviors of WOC and WOC-Hybrid subjected to elevated temperatures. The specimens were exposed to temperatures of 100–300 °C, and then the specimen was tested for tensile and compressive strength. The temperature 300 °C was used as the maximum elevated temperature due to the high water: cement ratio used in the mixed design (0.5W/C), which may cause damage to the furnace if we increase the temperature by more than 300 °C. To overcome this flaw, the application of machine learning models such as artificial neural networks (ANN) and multiple linear regression (MLR) was employed in this study to predict the compressive and tensile strength of plain concrete (PC), waste ceramic optimal concretes (WOC), and WOC-Hybrid fiber concrete (PVA–CR–WOC).

The ANN is a widely used and efficient tool for different types of analysis. The elevated temperature (100 °C, 200 °C, and 300 °C) is considered as input data and compressive strength (CS), tensile strength (TS) of PC, WOC, and WOC-Hybrid concrete used as target data. The Levenberg–Marquardt training function is used to train the target data.

Overall, most studies on ceramic waste materials have focused on mechanical properties at ambient temperature; however, some characteristics have not yet been clarified, including the mechanical behaviors under extreme conditions such as exposure to fire or high temperatures. Therefore, further experimental and computational studies on the fire performance of ceramic waste materials with and without fibers are necessary to promote its use. Thus, this study’s main contributions are: (a) to evaluate the potential use of (ceramic powder, fine and coarse ceramic aggregate) and hybrid fiber (PVA and CR) on the mechanical properties, crack width, and propagation of concrete subjected to elevated temperatures; (b) to provide a basis for further research on WOC and WOC-Hybrid concrete and its potential applications, and; (c) to compare the performances of ANN and MLR for evaluation and prediction the mechanical properties of PC, WOC, and WOC-Hybrid concrete.

## 2. Research Significance

The present study aims to develop a low-cost method for increasing concrete resistance at elevated temperatures. Due to hybrid fibers and ceramics’ advantageous properties, the simultaneous use of these two materials in concrete mixes reduces concrete costs and environmental pollution and enhances mechanical properties when exposed to high temperatures. Furthermore, since the main reason for concrete collapses under elevated temperatures is limestone, natural materials may be partially or even entirely replaced by ceramic material to prevent such damage to concrete damage. Furthermore, to prevent concrete cracking, reduce internal stresses, and increase its thermal conductivity, hybrid fibers have been used. This study also assesses the ability of machine learning models such as artificial neural networks and multiple linear regression (MLR) to predict the strength (compressive and tensile) of three types of concrete (PC, WOC, WOC-Hybrid). A MATLAB program has been created for this purpose, and representative models have been run to demonstrate efficiency and correctness. In addition to the estimate of concrete mechanical characteristics, the suggested neural system has been designed. By supplying all of the necessary information, anybody may test the proposed model.

## 3. Experimental Analysis

### 3.1. Description of Materials Used

In the experimental work, locally available materials such as ceramic waste powder, coarse ceramic aggregate, and fine ceramic aggregate have been used to replace cement (OPC 43 grade), natural sand, and natural coarse aggregate, respectively (Figure 1 and Figure 2). Waste ceramic floor tiles were collected from ceramic stores in Aligarh, which were cleaned and freed from dust then crushed by hammer into different sizes: 20 mm and 10 mm (waste ceramic aggregate—A_WC_); 4.75 mm (waste ceramic sand—S_WC_), and; 75 μm (waste ceramic cement—C_WC_), shown below in Figure 3 and Figure 4. The various properties are shown in Table 1 and Table 2. Types of fiber-reinforcements have been used, namely, crimped steel fiber (CR) and polyvinyl alcohol Fiber (PVA), shown in Figure 5, where its properties are shown in Table 3.

### 3.2. Preparation of Specimens

Overall, 21 cylindrical specimens with a diameter of 150 mm and a height of 300 mm were cast in three groups for testing under high temperatures, as shown in Figure 6, whereas groups are shown in Table 4.

For the present study’s purposes, 21 cylindrical specimens, each 150 mm in diameter and 300 mm in height, were cast in three groups, as shown in Table 3.

First Group: on the basis of test methods in accordance with IS 10262 guidelines [24], Plain Concrete (PC) was prepared with natural aggregate (coarse and fine) and Ordinary Portland Cement (OPC 43 grade) with a 0.5 water: cement ratio to be heated in a furnace at 100 °C, 200 °C, and 300 °C.

Second Group: on the basis of experimental results [8], waste ceramic optimal concrete (WOC) was obtained by replacing natural coarse aggregate with 20% ceramic aggregate, fine natural aggregate with 10% ceramic sand, and 43-grade OPC with 10% ceramic powder to be tested at 100 °C, 200 °C,300 °C.

Third Group: on the basis of experimental results [25], WOC-Hybrid was obtained by adding 1% PVA fiber and 1% CR-steel fiber with waste ceramic optimal concrete (WOC) to be heated to 100 °C, 200 °C, and 300 °C in a furnace. Designations for various types of concrete are shown in Table 5.

### 3.3. Concrete Mix Design

The concrete mixture for reference specimens containing natural aggregate and ordinary Portland cement (43-grade) is designed for compressive strength of 25 MPa. The water quantity (190 kg/m^3^) and W/C ratio (0.5) were the same for all the concrete mixes. However, they differed in their cement and aggregate (coarse and fine) content because waste ceramic was used as a partial replacement for either cement, or coarse or fine aggregates. The quantities of ceramics material, natural material, and fibers used are reported in Table 6. A_WC_, S_WC_, and C_WC_ represent coarse ceramic aggregate, fine ceramic aggregate, and ceramic powder weight per cubic meter of concrete. The mix proportions for different types of concrete are shown in Table 6.

### 3.4. Test Procedures

The most common method to elevate concrete fire resistance is by heating the furnace’s specimens for 28 days. In the current study, the specimens were heated in normal conditions at a high temperature. The high-temperature electric furnace (Figure 7) was fabricated to test concrete specimens at an elevated temperature of 1150 °C [26]. The inner size of the furnace is (1000 mm × 760 mm× 510 mm). Refractory coating was applied to all six sides of the furnace, with the heating elements attached to the left and right sides and the furnace’s top.

The compressive and tensile strengths of PC, WOC, and WOC-Hybrid concrete was determined using the 21 cylinders with 150 mm × 300 mm (diameter × height). After 24 h, these specimens were demolded and cured for 28 days. The specimens were prepared for testing at a single heating–cooling cycle from ambient temperature to an elevated temperature ranging from 100 °C to 300 °C at an interval of 100 °C after the submerged curing process. The researchers used 300 °C as the maximum limit of elevated temperature due to the high water: cement ratio used in the mixed design (0.5 W/C), and damage to the furnace would occur if we increase the temperature to more than 300 °C.

Automatically after heating up to the target temperature (100 °C, 200 °C, and 300 °C) at an average rate of 5 °C/min, (ramp rate 1–2 h), the temperatures were on hold for 1 h and then it was automatically closed down, which is shown in Figure 7b, after the cooling process stander ramp downtime of 20 h. All cylindrical specimens’ compressive and tensile strength were observed at ambient temperatures and elevated temperatures after testing the specimens under the testing machine.

### 3.5. Experimental Analysis Based on Laboratory Model

#### 3.5.1. Effect of Elevated Temperatures on Concrete Compressive Strength

The compressive strength of unheated concrete samples is comparable to that of heated concrete samples and oscillates around 25 MPa (values from 24.64, 27.99, to 28.8 Mpa for WOC, PC, and WOC-Hybrid, respectively). At ambient temperature (25 °C), ceramic material can increase the compressive strength by up to 39.82% for 10% ceramic powder replacement with OPC (43-grade). This finding illustrates that ceramic powder had higher strength than cement, which increased in concrete compressive strength (7 days’ compressive strength of OPC = 21.1 Mpa, 7 days’ compressive strength of C_WC_ = 37 Mpa) [8].

Hybrid fibers (PVA–CR) can improve the compressive strength of concrete, especially when fiber distribution is uniform in concrete. Using waste ceramic optimal concrete (WOC) with hybrid fibers effectively increased the compressive strength by up to 16.87% [25]. The compressive strength test results obtained at elevated temperatures are plotted in Figure 8.

First Group: the CS decrement for PC concrete under 100 °C, 200 °C, and 300 °C is 11.61%, 18.19%, and 36.01%, respectively. Chang, Y.F. et al. [27] have confirmed this performance, where they studied concrete compressive strength after exposure to high temperatures. Based on the comparison between the literature and the present study, it was observed that the decrease in CS of PC under elevated temperature (200 °C) was 17% [27]. In contrast, CS decrement in the present study is 18.19%. The results validated the past results.

Second Group: the CS decrement for WOC concrete under 100 °C, 200 °C, and 300 °C is 8.48%, 16.72%, and 34.25%, respectively. Up to a temperature of 200 °C, the mixtures with a ceramic material (WOC) show a smaller reduction in the compressive strength compared to the reference mixture. Netinger, I. et al. [4] reported similar behavior on the addition of waste ceramic tiles in concrete under elevated temperature; based on the comparison between previous studies and the present study, it is observed that the decrement in CS under elevated temperature (200 °C) in specimens containing ceramic aggregate was only 4.43% [4]. In contrast, specimens containing ceramic powder and ceramic aggregate (fine and coarse) are 8.48% (WOC). This concludes that decrement in CS of specimens containing ceramic aggregate is less than decrement in CS of specimens containing both ingredients (aggregate + cement).

Third Group: hybrid concrete shows good performance under elevated temperature compared to reference concrete, where the reduction in CS for WOC-Hybrid concrete is 7.15%, 14.55%, and 19.51%, respectively.

Hager, I. [28] concluded that the following two stages characterize the compressive strength–temperature relationships of concrete: the stage of minor strength loss occurs between room temperature and approximately 100 °C for all types of concrete. The permanent stage of strength loss begins around 200 °C for both PC and WOC concrete. All failure specimens after exposure to elevated temperatures of 100 °C, 200 °C, and 300 °C under compressive strength machine test have been shown in Figure 9.

#### 3.5.2. Effect of Elevated Temperatures on Concrete Tensile Strength

Concrete behavior in tension, whether directly or indirectly, has received scant attention. Consequently, limited research has been conducted in this area. Furthermore, most of the TS tests found in the literature are for high-performance concrete.

Tensile strength decreases with increasing temperature, as illustrated in Figure 10. However, according to Chang, Y.F. et al. [27], the decrease in tensile strength is greater than the decrease in compressive strength for the same heat treatment.

At ordinary temperature (25 °C), hybrid fiber (PVA–CR) was found capable of improving the tensile strength (TS) of concrete, especially when well distributed in concrete. Using waste ceramic optimal concrete (WOC) together with hybrid fibers (PVA + CR) was shown to be effective in increasing the TS by up to 85.44% [25].

First Group: the TS decrement for PC concrete under 200 °C and 300 °C is 19.21% and 35.91%, respectively. The decrease in TS could be due to the decrease in cohesive forces associated with water evaporation at this temperature. The reduction may also be due to shrinkage caused by free water evaporation and chemically bound water in cement paste, weakening the bond between aggregate–cement pastes. Chang, Y.F. et al. [27] have reported more or less similar PC tensile strength behavior under elevated temperatures. Based on the comparison between previous studies and the present study, it is observed that the decrease in TS of PC concrete under an elevated temperature of 200 °C was 20% [27], and the reduction in TS in the present study is 19.21%. Thus, the previous results are validated.

Second Group: the WOC specimens behave better than PC specimens with the increasing temperature, and the TS decrement for WOC concrete at 200 °C and 300 °C was 14.18% and 30.6%, respectively. For tensile strength, the mixture with ceramic material was more fire-resistant than the reference mixture over the whole temperature range. The concrete with ceramic tiles has lower thermal conductivity than concrete with natural aggregates, which might explain such mixtures’ better fire resistance in this research.

Third Group: hybrid concrete shows good performance under elevated temperature compared to WOC, where the reduction in TS for WOC-Hybrid concrete under 200 °C and 300 °C is 10.26% and 17.10%, respectively. The current tests demonstrate that the addition of hybrid fibers can improve the mechanical properties of concrete at elevated temperatures, especially when tensile strength is considered. Such improvements could be attributed to the mix of two types of fibers such as metallic and non-metallic, contributing to resisting normal and elevated temperatures. Figure 11 shows PC, WOC, and WOC-Hybrid cylinders tested under a tensile strength machine at failure after exposure to elevated temperatures 100 °C, 200 °C, and 300 °C.

#### 3.5.3. Effect of Elevated Temperatures on Behavior of New Generation Concrete (WOC and WOC-Hybrid Concrete)

Mechanical properties of concrete specimens were decreased by increasing temperature, as shown in Table 7. This decrease reached 30% in comparison to unheated specimens. The addition of waste ceramic materials and hybrid fibers in concrete enhances the strength in concrete at high temperatures.

According to Netinger, I. et al. [4], concrete with ceramic tiles has lower thermal conductivity than concrete with natural aggregates, which might explain WOC mixtures’ better fire resistance in this research. In addition to using ceramic material to improve heat resistance, hybrid fibers (metallic and non-metallic) and ceramic concrete show very high resistance under elevated temperatures (below 400 °C). Thus, the concurrent use of these two materials in concrete appears to reduce environmental pollution and cost, and enhance the mechanical properties of concrete exposed to high temperatures.

Generally, the addition of non-metallic fibers (PP or PVA) has no significant effect on improving concrete’s CS after exposure to elevated temperature. However, such improvement is visible to a certain extent when FS and TS are considered, particularly at a temperature below 400 °C. Non-metallic fibers (PP or PVA) can increase concrete resistance to cracking, improving its behavior under the TS test (Table 7). However, Peng, G.F. et al. [29] prove that “the melting and ignition points of non-metallic fiber are around 150–400 °C. Thus, the strength of fiber-reinforced concrete decreases when the temperature is above 400 °C due to melting up non-metallic fibers leaving the pores acting at a disadvantage for concrete under any test”.

The addition of metallic fibers (HK or CR) can improve the concrete mechanical properties at elevated temperatures when CS and TS are considered. These improvements may be a result of the fact that the testing temperatures are not high enough to melt steel fiber. Furthermore, Gao, D. et al. [12] concluded that “metallic fiber has higher thermal conductivity than normal concrete materials such as aggregate and cement. Consequently, heat can transmit more uniformly in the fiber-reinforced concrete to reduce the cracks caused by a thermal gradient in concrete, improving concrete performance under compressive and tensile strength tests. Lastly, the resistance to elevated temperatures provided by metallic fiber is weaker than that provided by non-metallic fiber because of the reduced thermal gradient.

#### 3.5.4. Effect of Elevated Temperatures on Concrete Crack Width and Propagation

Crack formation, width, and propagation occur in concrete structures at high temperatures. This effect can make the structure unusable and a risk to life and property by causing death or permanent damage. Cracks in concrete specimens under high temperatures caused a decline in unit weights.

Demir, A. [30], Tang, Y. et al. [31], Sancak, E. and Şimşek, O., [32], and Alp, I et al. [33] have studied concrete crack width and propagation as well as the effect of high temperature on the changing of mechanical properties of concrete. In these studies, industrial waste was used in concrete technology such as tile waste and crushed ceramic waste.

After testing, the specimens were checked for crack width and propagation, since that was one of the purposes of this research. High-resolution digital pictures of the cylinder were taken and imported into specialized software for manual tracing of the crack pattern for each specimen. At the same time, the crack width was measured using a mobile laser crack width reader with a reading range from 0.03 mm to 2.5 mm, with an increment of 0.05 mm.

Figure 12 shows the crack pattern for all the specimens that showed such cracking. It can be noted that for plain concrete (PC) and waste ceramic concrete (WOC) the number of cracks is higher than is the case with hybrid fibers (WOC-Hybrid).

Previous studies noted that the damage and cracks mainly accumulate and develop at the interface between the waste materials particles and the cement matrix [34,35].

The non-metallic fibers reinforced concrete has much better resistance to thermal spalling compared to the concrete without fibers because of the melting and ignition of non-metallic fibers. Consequently, the pores formed to expand to form micro-cracks, connecting the existing capillary pores to provide channels for water vapour to escape. Therefore, an optimum dosage of non-metallic fibers (PVA or PP) around 0.5–1% by the mix’s volume is recommended for concrete to obtain high-temperature crack resistance [36].

## 4. Machine Learning Models

### 4.1. Concept of Machine Learning Model

Neurons are the structural and functional components of the nervous system that relay information between the brain and the body. A neuron has its biological structure composed of (a) the cell body, (b) the axon, and (c) the Dendron.

ANN and MLR are a type of machine learning technique that depends on biological neuron networks as a foundation. This is the anatomy of biological and artificial neural networks, as depicted in Figure 13.

ANN is built out of multiple hidden layers and an output layer. The single-layer network has a source node as an input layer and a neuron as an output layer. One or more hidden layers of neurons build out the multilayer network. When it comes to output accuracy, the number of neurons in the input plays a significant role.

The three essential aspects of ANN have been described by Paral, A. et al. [37] as follows:(a)The arrangement of connections between neurons is known as network architecture.(b)The weights of the links are determined using a learning method.(c)The activation function, which is a neuron’s intrinsic state. A single input or multiple input system is used to carry out this training procedure. It may also be divided into single and multiple hidden layers (Figure 13).

One of the most basic mathematical models is the multiple linear regression model (MLR). It works based on linear connections between inputs and outcomes. In other words, it involves a constant regression in the formula to extract the linear correlations between dependent and independent variables [38,39]. Even if the mathematical concept of this method is relatively simple compared to other machine learning models, it has proved its efficiency in solving several engineering problems [40].

### 4.2. The Architecture of the Machine Learning Model

#### 4.2.1. Artificial Neural Network (ANN)

There are three distinct architectures for ANN networks: single-layer feed-forward networks, multilayer feed-forward networks, and recurrent networks. This study examined the feed-forward network with a single hidden layer. This single neuron system carries out three different functional activities. Demuth and Beale’s single and multiple input neuron systems architecture is illustrated [41] (Figure 14).

Initially, the term *p* is a scaler unit, multiplying with scalar weight *W*, and forms a product *W_P_*. Next, the weighted input *W_P_* is added with scalar bias *b* to develop a net input *n.* Finally, the net input *n* crosses through the transfer function *f,* which forms a scalar output *a*. The output of the single-input system is shown in Equation (1):(1)$$a=f\left(W_{p}+b\right)$$

The weight function, net-input function, and transfer function are the names for these three processes. The transfer or activation function *f* that is chosen can significantly impact the ANN’s complexity and performance. Although sigmoidal transfer functions are the most popular, other types of functions can be employed. A significant variety of potential transfer functions have been presented in previous research [42]. The logistic sigmoid function was determined to be appropriate for the problem examined in this work, as shown in Figure 15.
(2)$$n=W\cdot P=W_{i,1}P_{1}+W_{i,2}P_{2}+\ldots +W_{i,R}P_{R}+b\;$$

The expression of a single neuron can be written in matrix form, as in Equation (3).
(3)$$n=W_{p}+b\;$$

Therefore, the neuron output is as written in Equation (4).
(4)$$a=f\left(W_{p}+b\right)\;$$

For $R_{i}$ number of input and *Si* number of neurons, the weighted matrix is written as in Equation (5).
(5)$$\begin{array}{ccc} W_{1,1} & W_{1,2} & W_{1,Ri}\\ W_{1,2} & W_{2,2\;} & W_{2,Ri}\\ W_{si,1} & W_{si,2} & W_{si,Ri} \end{array}\;$$

Since the training method used for optimization is essential in creating quality mapping, a thorough analysis was carried out to determine the best method for this problem. As a result, the back-propagation technique is the most often utilized approach in the literature.

#### 4.2.2. Multi-Linear Regression

MLR work is based on the equation below:(6)$$y=b_{0}+b_{1}x_{1}+b_{2}x_{2}+\ldots b_{i}x_{i}$$
where:*y*: the independent variable.*b*: the regression constant.*x*: the *i*th predictor.

### 4.3. Computational Model Development

This section presents the process for tuning optimum ANNs and MLR used for the training, validation, and test of the mechanical properties of plain concrete (PC), waste ceramic optimal concrete (WOC), and hybrid concrete (WOC-Hybrid) based on experimental data (Section 3).

#### 4.3.1. Model’s Performance Criteria

The correction coefficients and the new specific terms must consider the ceramic concrete mechanical properties and replacement ratio of ceramic materials. The proposed expressions and the correction coefficients have been analyzed using formulas statistical indexes: (a) the Pearson correlation coefficient and (b) the mean squared error (MSE). Lastly, the best prediction expressions for the compressive and tensile strength of structural PC, WOC, and WOC-Hybrid are proposed.

#### 4.3.2. Input and Output parameters

##### ANN Model

Water (W), natural coarse aggregate (NCA), waste ceramic aggregate (A_WC_), ordinary Portland cement (OPC), waste ceramic cement (C_WC_), natural fine aggregate (NFA), waste ceramic sand (S_WC_), crimped fiber (CR), and polyvinyl alcohol (PVA fiber) are the appropriate input training vectors of dimension 10 × 12 for compressive strength training and 10 × 9 for tensile strength training. Compressive strength (CS) and tensile strength (TS) are the appropriate output training vectors of dimension 10 × 1 for compressive strength training and 10 × 1 for tensile strength training. Table 8 lists their values, as well as the highest and lowest values. 

The NNTOOL command was used to train the ANN model in this work using the MATLAB package. The number of connections and the neural network topologies (which vary in complexity and efficiency) determine neural networks’ power. Figure 16 depicts the neural network design for a single layer as well as the network flow diagram. Due to its ease of usage, the feed-forward neural network was chosen in the present research. The input data proceed in a single path and are routed through artificial neural nodes to the output nodes in this approach. The number of layers depends on the complexity of the function [42]. There are 10 input data and 2 output data in this study. As a result, the feed-forward neural network is a good fit for this research.

ANN was trained using the Levenberg–Marquardt back-propagation algorithm. Figure 17 depicts an ANN composed of an input layer with 10 neurons, a hidden layer with four trial neurons 3, 5, 7 and 10, respectively, and an output layer with one neuron.

Many different ANN models were built in this study to evaluate the sensitivity of the ANN outcomes. Each of these ANN models was trained using 60% of the data points, and the trained ANN was validated and tested using the remaining 40%. In particular, 20% of the data points were utilized to validate the trained ANN and 20% of the data points were utilized for testing.

Different ANN models were constructed and studied based on the above to determine the best ANN model for predicting the compressive and tensile strength of PC, WOC, and WOC-Hybrid concrete. Based on Pearson’s correlation coefficient values, the created ANN models were ranked in decreasing order. The optimal ANN for the PC, WOC, and WOC-Hybrid mechanical characteristics is shown in Figure 18.

For the PC, WOC, and WOC-Hybrid mechanical characteristics, Figure 18 and Figure 19 show the mechanical characteristics (CS and TS) of the precise experimental values vs. the predicted values of the optimal ANN model. The 28-day compressive strength and tensile strength of concrete material predicted using the NNTOOL feed-forward neural network are very close to the experimental results.

##### MLR Model

In the present paper, the MLR model was applied to predict two concrete characteristics, tensile strength and Compressive strength. A total of nine samples were used to predict the tensile strength (77% for training and 23% for test), and 12 samples were used to predict the compressive strength (75% for training and 25% for test).

### 4.4. Machine Learning Model Results

Due to the system’s complexity, it is difficult to determine the effect of a single parameter change on the model’s various output parameters. However, a parametric analysis was carried out to check the ANN and MLR models’ capability to capture the sensitivity of PC, WOC, and WOC-Hybrid concrete mix properties to individual constituents (Table 9).

#### 4.4.1. ANN Model

##### Tensile Strength Model

The artificial neural network modeling results for all input parameters and all PC, WOC, and WOC-Hybrid models with a different number of hidden neurons indicated that elevated temperatures have the greatest effect on the TS of concrete (Figure 20).

Figure 19 illustrates the simulation results for tensile strength after 28 days for PC, WOC, and WOC-Hybrid concrete. In this case, it was discovered that elevated temperatures had a significant effect on the output parameters. According to this analysis, all PC, WOC, and WOC-Hybrid concrete mixes meet the required tensile strengths. Tensile strength decreases as elevated temperature increases at any elevated temperature level (Figure 20). All predicted TS values obtained by the ANN model for all temperatures in this analysis are within the range which is nearly identical to other experimental studies conducted by several researchers, [43,44]. The tensile strength of concrete had remained within an acceptable range (as per IS code).

##### Compressive Strength Model

Artificial neural network modeling was used to determine the effect of elevated temperatures on the CS of concrete for all input parameters and all PC, WOC, and WOC-Hybrid models with a different number of hidden neurons (Figure 21).

Figure 22 illustrate the simulation results for compressive strength after 28 days of PC, WOC, and WOC-Hybrid concrete. In this case, it was discovered that elevated temperatures had a significant effect on the output parameters. The results of this analysis indicate that all PC, WOC, and WOC-Hybrid concrete mixes meet the minimum compressive strength requirements. Compressive strength decreases as elevated temperature increases at any elevated temperature level (Figure 22). The predicted CS values obtained by the ANN model for all temperatures in this analysis all fall within the range. Thus, the compressive strength of all concrete types remained within an acceptable range.

##### ANN Model Performance

The performance of the developed models by ANN can be drawn from the Figure 16, Figure 17 and Figure 18 mentioned above:(a)Among all available training algorithms in the literature, the Levenberg–Marquardt algorithm produced the best ANN prediction of WOC strength.(b)It was determined that the optimal number of hidden layers for the top ten models is 1.(c)Initial weights significantly affect the results; varying the initial weights results in varying the optimal ANN architectures.

#### 4.4.2. MLR Model

##### Tensile Strength Model

According to Table 10, the building of the MLR model to predict tensile strength was mainly based on three parameters: coarse aggregate, CR fiber, and temperature, with linear coefficients equal 0.003, 0.014, and −0.004, respectively. It should mention that the model has a linear constant equal to −0.045.

Based on the following results, the mathematical model to predict the tensile strength has the following form:Tensile strength = −0.045 + Coarse aggregate × 0.003 + CR fiber × 0.014 − Temperature × 0.004

Figure 23 presents a scatter plot between the actual (X axis) and predicted (Y axis) value of tensile strength. The red line presents the 1:1 line, which is the total congruence between the two axes. The blue circles present the samples used in the training phase, whereas the green circles present the samples used in the test phase. According to Figure 20, it is clear that the model has high efficiency in predicting the tensile strength where the R^2^ achieves 0.9862 in the training phase and 0.9983 in the test phase.

##### Compressive Strength Model

Table 10 presents the coefficients of the developed MLR model for the prediction of compressive strength. The mathematical model has a linear constant equal to 16.45, where the model is based mainly on coarse aggregate, ceramic sand, CR fiber, and temperature for the prediction of compressive strength. The coefficients of the main predictors were 0.0102, 0.0004, 0.00255, and −0.0288, respectively. These values give the final model the following form:Compressive strength = 16.45 + Coarse aggregate × 0.0102 + ceramic sand × 0.0004 + CR fiber × 0.0255 − Temperature × 0.0288

As with Figure 23, Figure 24 consists of the same components, but this time to present the results of predicting the compressive strength using the developed MLR model. The model shows high efficiency in predicting the compressive strength, where the R^2^ achieved 0.9715 in the training phase, and 0.9687 in the test phase.

##### MLR Model Performance

The performance of the developed models during training and test periods was assessed by calculating three performance criteria, namely, mean square error (MSE), root mean square error (RMSE), and mean absolute error (MAE). The formulas used for the calculation of the previous indicators is mentioned in Table 11.

Table 12 presents the calculation results of model performance during training and test phases, where the results shows that there is a slight augmentation in the values of performance indicators regarding the tensile strength model where the MSE, RMSE, and MAE were 0.1487, 0.3856 and 0.363 in the training phase and 0.2208, 0.4699, and 0.469, respectively. For the compressive strength model, the same observation has been registered, where the MSE, RMSE, and MAE had augmented from 0.3931, 0.6270, and 0.4739 in the training phase to 1.8889, 1.3744, and 1.2297 in the test phase.

## 5. Conclusions

The purpose of this study was to determine whether it was possible to increase the heat resistance of concrete exposed to high temperatures by developing a new concrete generation proposed by the authors (WOC concrete + WOC-Hybrid concrete), and to assess the ability of machine learning models (ANN and MLR) to predict the strength (compressive and tensile) of these types of concrete using a MATLAB program. The following conclusions can be drawn from the results of the experimental and computational test:(a)Experimental Analysis
The concrete compressive strength which contains natural material (PC) is decreased by up to 11.61%, 18.91%, and 36.01% when exposed to a temperature of 100 °C, 200 °C, and 300 °C. In comparison, this reduction is about 8.48%, 16.72%, and 34.25% in the specimens containing waste ceramic material (WOC). Thus, it may be concluded that ceramic material contributes to the heat resistance of concrete compared with natural material because concrete with ceramic tiles have lower thermal conductivity than concrete with natural aggregates.Due to the increase in the transport properties of heated material, concrete with hybrid fibers (PVA + CR) showed a good effect on concrete compressive strength under elevated temperatures. The reductions in compressive strength in specimens containing hybrid fiber are about 7.15%, 14.55%, and 19.51% when exposed to 100 °C, 200 °C, and 300 °C, respectively.The concrete tensile strength which contains natural material (PC) is decreased by up to 19.21% and 35.91% when heated to 200 °C and 300 °C. In comparison, this reduction is about 14.18% and 30.60% in the specimens containing waste ceramic material (WOC).Hybrid Fibers (PVA–CR) affected concrete tensile strength at elevated temperatures positively. This is because heat transmits more uniformly in the concrete reinforced with hybrid fibers and the cracks caused by a thermal gradient in concrete decrease, improving concrete behavior under tensile strength tests. The reductions in tensile strength in specimens containing hybrid fiber are about 10.26% and 17.10% when exposed to temperatures of 200 °C and 300 °C, respectively.Concrete crack width and propagation develop primarily at the interface between waste material particles and the cement matrix.
(b)Machine Learning Model Analysis
Machine Learning (ML) models such as ANN and MLR can evaluate concrete’s mechanical properties. It was demonstrated that the developed ML model was successfully trained, validated, and tested.Concrete is a complex material, and there is a correlation between many factors. ML is a highly interconnected system that can learn the nature of complex interrelationships between independent and dependent variables. As a result, the models could evaluate the compressive and tensile strength at 28 days.The ANN and MLR models’ capability to evaluate the effect of elevated temperatures between 100 °C and 1000 °C on the properties of PC, WOC, and WOC-Hybrid concrete was mentioned.The ML models were assessed using statistical tools, including R^2^ and MSE, and through visual assessment using scatter plots and line and bar diagrams. It has been found that best model performance between machine learning models in the training phase was registered for multiple linear regression (MLR), where the R^2^ value achieve achieved 0.9715 in the training phase and 0.9687 in the test phase.It would also be helpful to conduct studies on benchmarking of different prediction models. It is also suggested that ANN and MLR models be applied with different properties and materials parameters. In future work, we will use these techniques in different areas. We shall also discover possible enhancements to the method, such as adding different percentages of waste materials such as waste rubber and study their strength. Additionally, we shall encompass these approaches to the cooperative prediction of multiple parameters.


## Figures and Tables

**Figure 1 materials-15-02371-f001:**
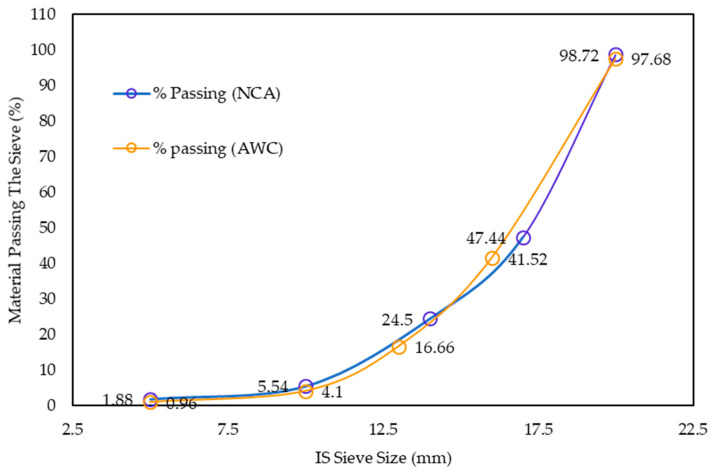
Particle size distribution of natural coarse aggregate and ceramic coarse aggregate.

**Figure 2 materials-15-02371-f002:**
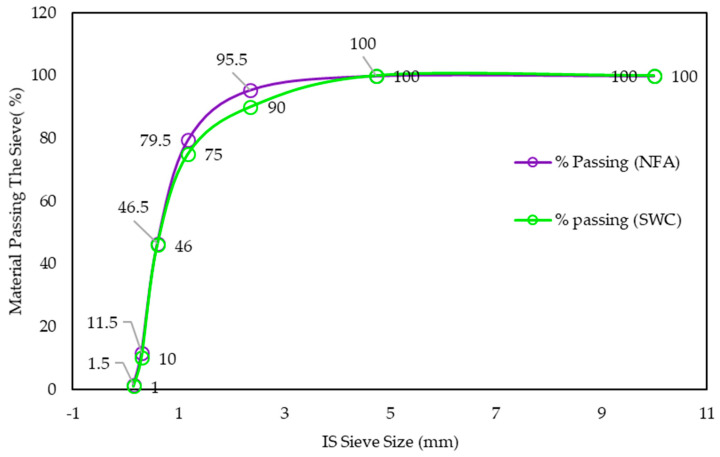
Particle size distribution of natural fine aggregate and ceramic fine aggregate.

**Figure 3 materials-15-02371-f003:**
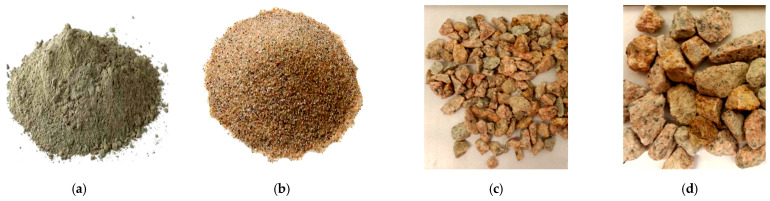
(**a**) commercial cement, (**b**) river sand, (**c**) natural coarse aggregates (NCA)—10 mm, (**d**) natural coarse aggregates (NCA)—20 mm.

**Figure 4 materials-15-02371-f004:**
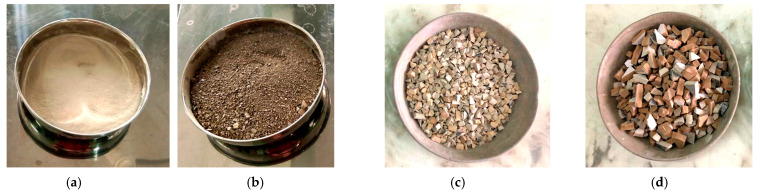
(**a**) waste ceramic cement (C_WC_)—75 µm, (**b**) waste ceramic sand (S_WC_)—4.75 mm, (**c**) waste ceramic aggregate (A_WC_)—10 mm, (**d**) waste ceramic aggregate (A_WC_)—20 mm.

**Figure 5 materials-15-02371-f005:**
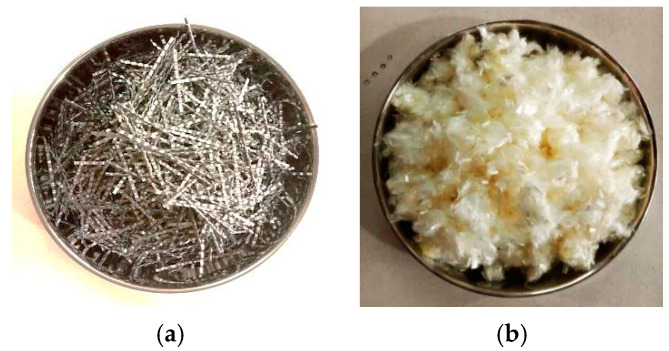
Metallic and non-metallic fiber: (**a**) crimped steel fiber (CR) (60 mm) (**b**) polyvinyl alcohol fiber (PVA) (12 mm).

**Figure 6 materials-15-02371-f006:**
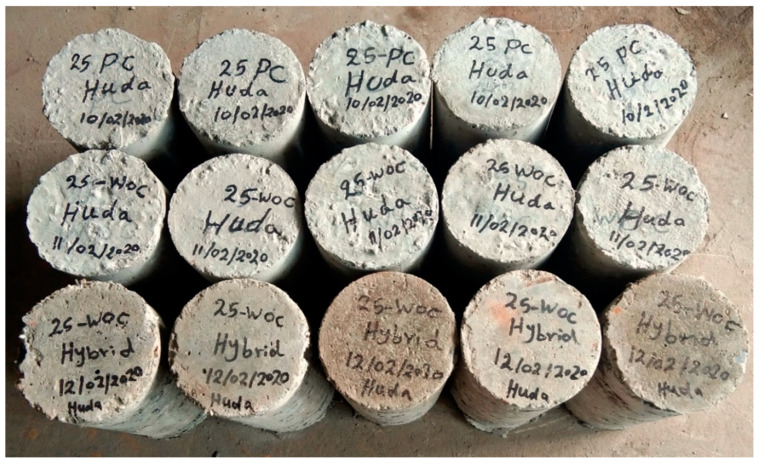
Prepare specimens for test.

**Figure 7 materials-15-02371-f007:**
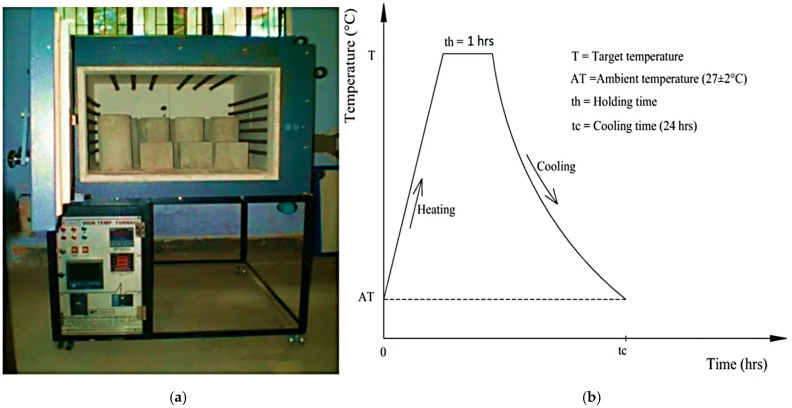
(**a**) High-temperature furnace, (**b**) single heating–cooling cycle curve.

**Figure 8 materials-15-02371-f008:**
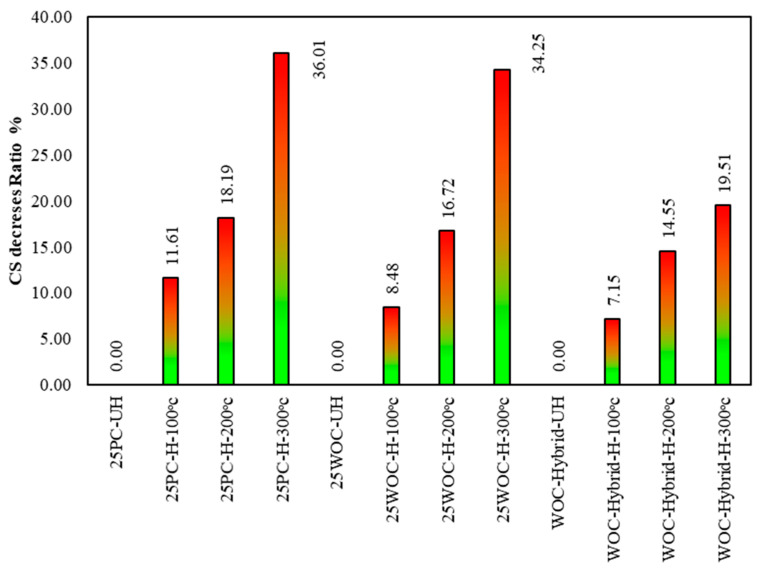
The percentage decrement in compressive strength of PC, WOC, and hybrid specimens at elevated temperatures to samples strength at 25 °C.

**Figure 9 materials-15-02371-f009:**
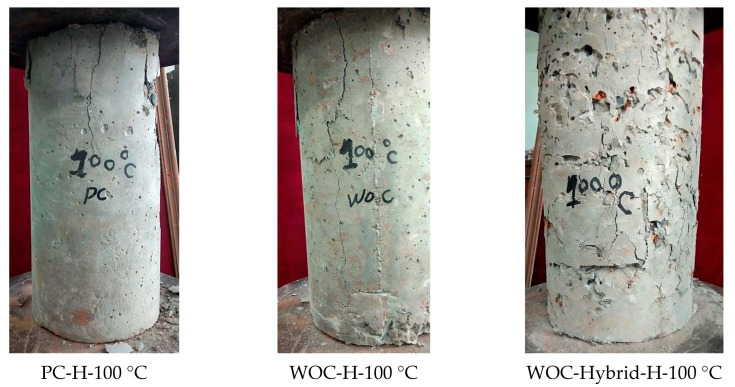
PC, WOC, and WOC-Hybrid cylinder under compressive strength machine at failure after exposure to elevated temperatures of 100 °C, 200 °C, and 300 °C.

**Figure 10 materials-15-02371-f010:**
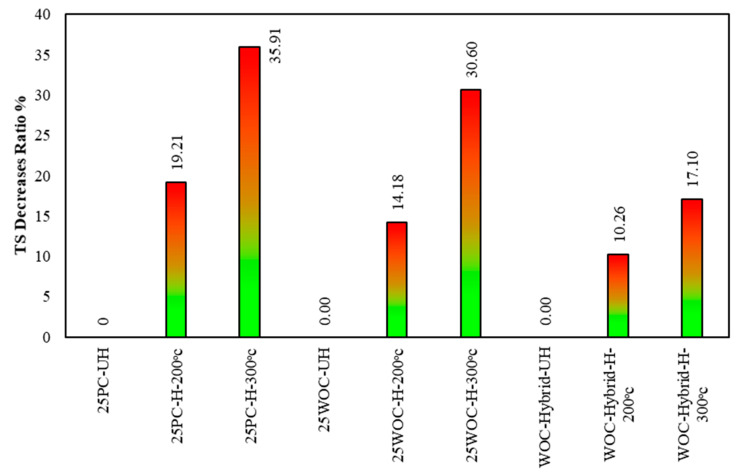
The percentage decrement in tensile strength of PC, WOC, and hybrid specimens at elevated temperatures to samples strength at 25 °C.

**Figure 11 materials-15-02371-f011:**
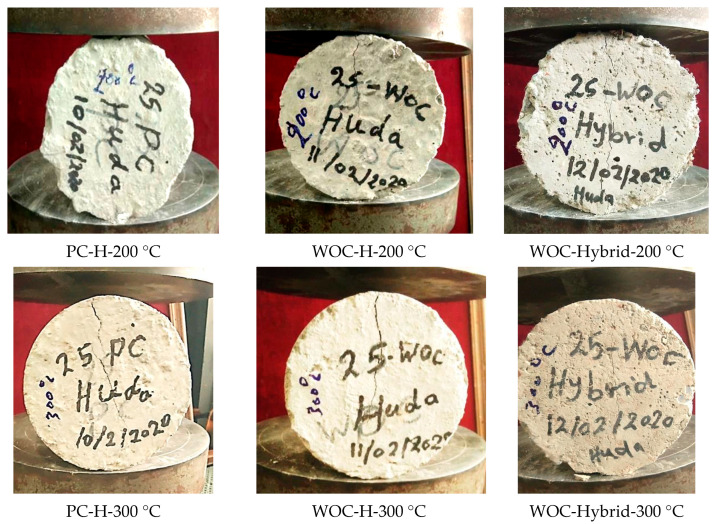
PC, WOC, and WOC-Hybrid cylinder under tensile strength machine at failure after exposure to elevated temperatures of 100 °C, 200 °C, and 300 °C.

**Figure 12 materials-15-02371-f012:**
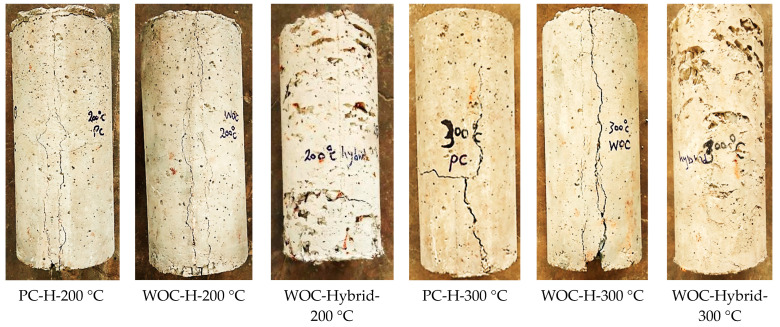
PC, WOC, and WOC-Hybrid cylinder at failure after exposure to elevated temperatures of 200 °C and 300 °C.

**Figure 13 materials-15-02371-f013:**
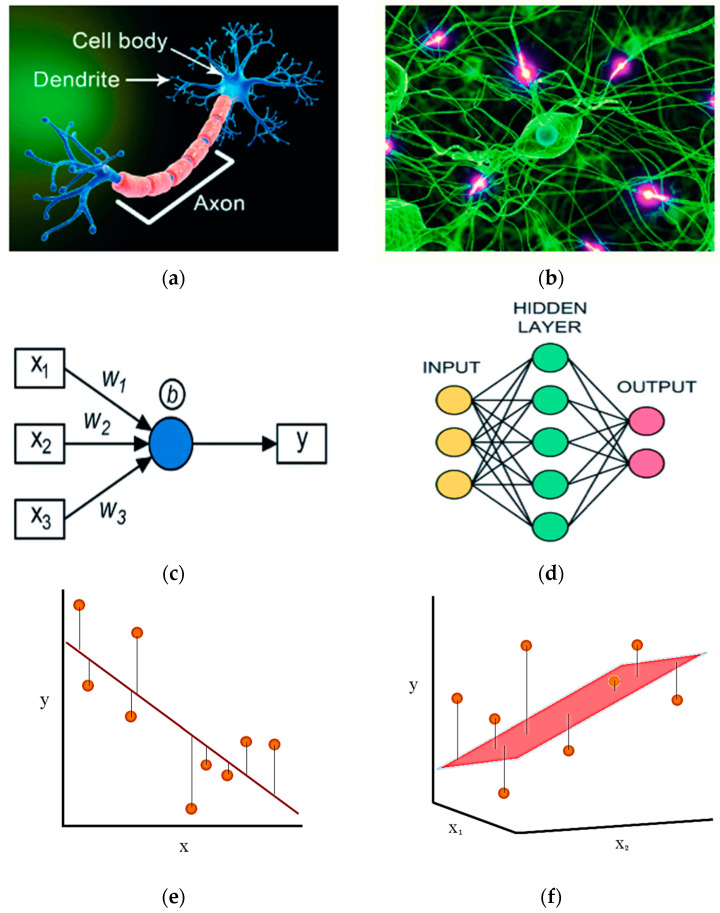
(**a**) Biological neuron, (**b**) biological neuron network, (**c**) artificial neuron, (**d**) artificial neural network, (**e**) simple linear regression, (**f**) multiple linear regression.

**Figure 14 materials-15-02371-f014:**
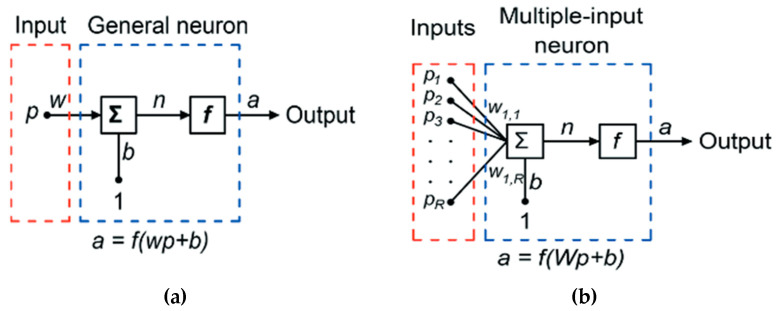
(**a**) Single-input neuron, (**b**) multiple-input neuron.

**Figure 15 materials-15-02371-f015:**
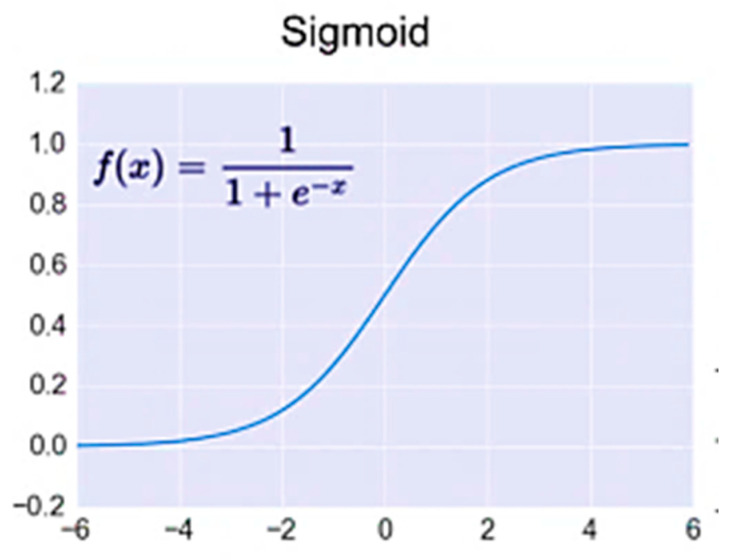
Activation function. Multiple input neurons have individual inputs *P*_1_, *P*_2_, *P*_3_....*P_RI_* weighted by *W*_1,1_, *W*_1,2_, *W*_1,3_....*W*_1,_*_RI_* and formed a weight matrix *W*. The neuron bias *b* summed with the weighted matrix and formed net input, *n*, shown in Equation (2).

**Figure 16 materials-15-02371-f016:**
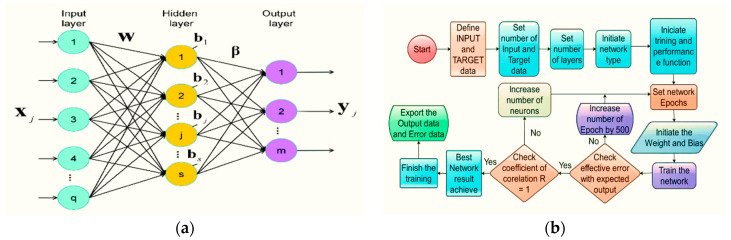
(**a**) Single hidden layer neural network architecture, (**b**) flowchart of the artificial neural network training process.

**Figure 17 materials-15-02371-f017:**
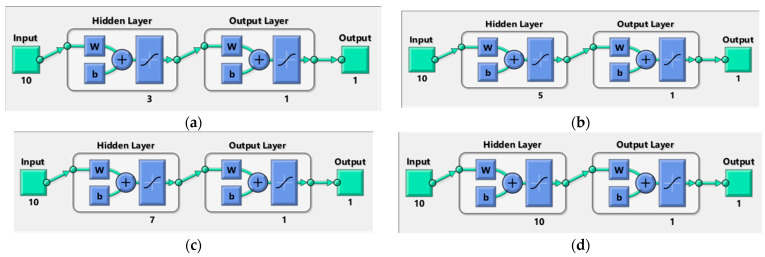
The architecture of ANN models for prediction compressive and tensile strength (**a**) 3 neurons, (**b**) 5 neurons, (**c**) 7 neurons and (**d**) 10 neurons.

**Figure 18 materials-15-02371-f018:**
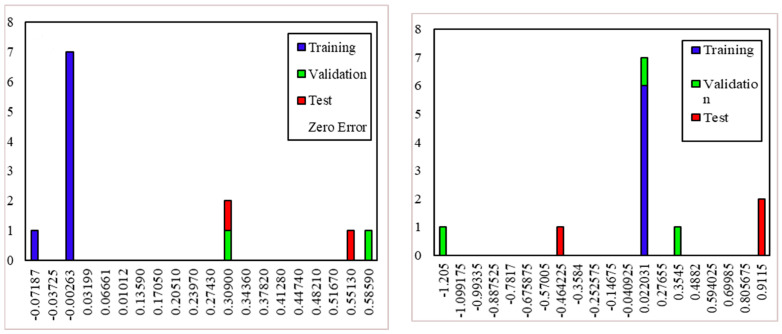
The Pearson’s correlation coefficient R and the mean squared error (MSE) of the experimental and predicted (**a**) CS for PC, WOC, and WOC-Hybrid concrete (**b**) TS for PC, WOC, and WOC-Hybrid concrete.

**Figure 19 materials-15-02371-f019:**
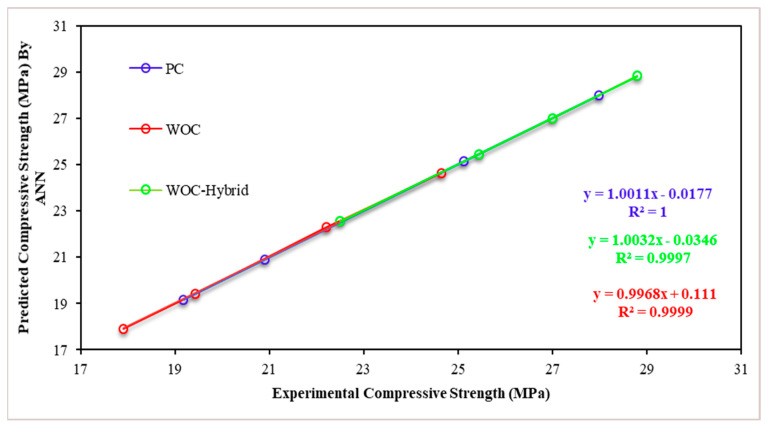
Linear relationship between measured and predicted compressive strengths (the Levenberg–Marquardt back-propagation algorithm).

**Figure 20 materials-15-02371-f020:**
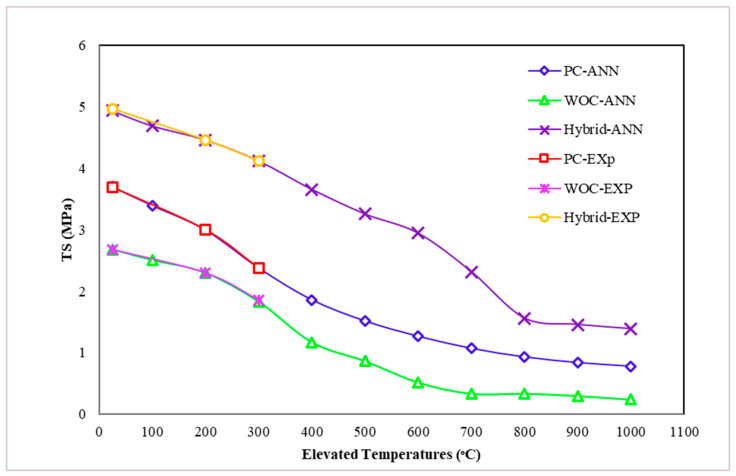
Effect of elevated temperatures on concrete tensile strength at 28 days (MPa).

**Figure 21 materials-15-02371-f021:**
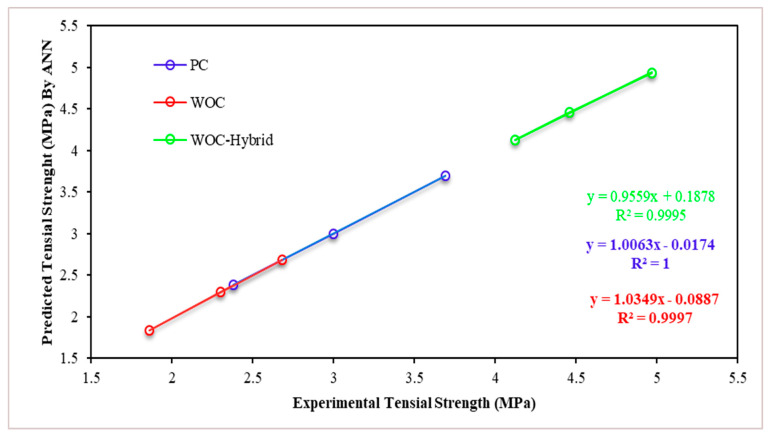
Linear relationship between measured and predicted Tensile strengths (the Levenberg–Marquardt back-propagation algorithm).

**Figure 22 materials-15-02371-f022:**
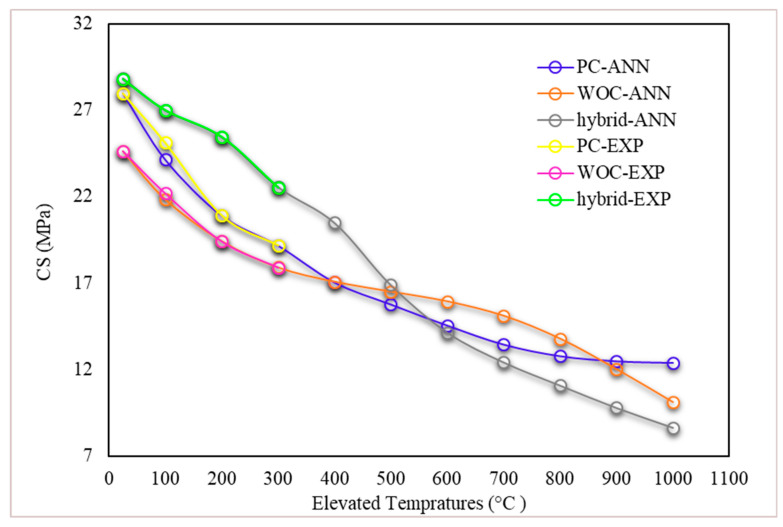
Effect of Elevated Temperatures on Concrete Compressive Strength at 28 Days (MPa).

**Figure 23 materials-15-02371-f023:**
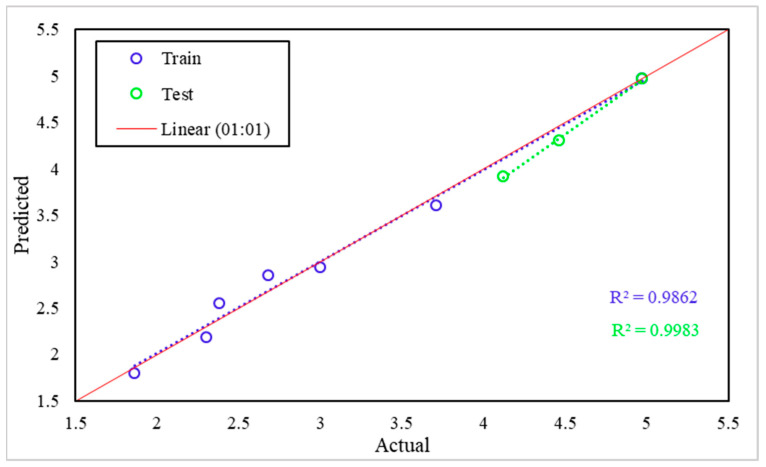
Tensile strength scatter plot between actual and predicted values.

**Figure 24 materials-15-02371-f024:**
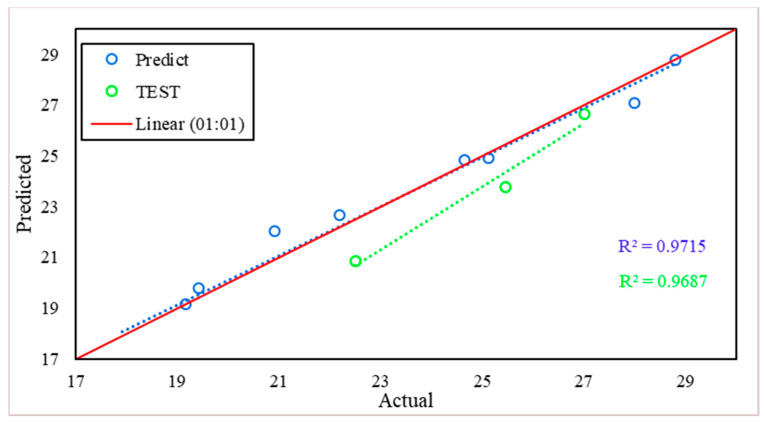
Compressive strength scatter plot between actual and predicted values.

**Table 1 materials-15-02371-t001:** Properties of used materials.

Physical Properties	Cement– OPC	NCA	Sand	C_WC_	A_WC_	S_WC_
Normal Consistency (%)	32	-	-	8	-	-
Specific Gravity	3.15	2.84	2.64	2	2.31	2.26
Initial Setting Time	42 min	-	-	54 min	-	-
Final Setting Time	600 min	-	-	680 min	-	-
7 Days’ Compressive Strength	21.1 MPa	-	-	37	-	-
Fineness Modules	-	6.99	2.65	-	6.98	2.2
Maximum Size	75 µm	0.02 m	4.75 mm	75 µm	0.02 m	4.75 mm
Density (kg/m^3^)	1440	1550	1650	-	-	-
Water Absorption (%)	-	0.23	2.24	-	0.55	2.52
Crushing Value (%)	-	34	-	-	20.86	-
Impact Value (%)	-	24	-	-	27	-

NCA: Natural Coarse Aggregate; A_WC_: Waste Ceramic Aggregate; S_WC_: Waste Ceramic Sand; C_WC_: Waste Ceramic Cement.

**Table 2 materials-15-02371-t002:** Chemical analyses of waste ceramic powder and OPC.

Materials	Waste Ceramic Powder (C_WC_)	Cement (OPC 43)
SiO_2_	68.85	22.18
Al_2_O_3_	17	7.35
Fe_2_O_3_	0.8	3.83
CaO	1.7	63.71
Na_2_O	-	0.28
K_2_O	1.63	0.11
MgO	2.5	0.95
TiO_2_	0.737	0.13
MnO	0.078	0.04
LOI	1.78	1.6

**Table 3 materials-15-02371-t003:** Properties of used fibers.

Fiber Type	CR	PVA
Surface	Plane	-
Cross-section	Circular	-
Anchorage	Continuous	Straight
Length (mm)	50 mm	12 mm
Diameter (mm)	1	0.04
Aspect ratio	50	300
Density (g/cm^3^)	7.85	1.3
Tensile strength (MPa)	1250	1560
Elastic modulus (GPa)	200	41

**Table 4 materials-15-02371-t004:** Description of group testing of total 21 test specimens.

Group	Samples	Purpose of Casting			
		CS (Cylinder)		TS (Cylinder)	
		N.	Size (mm)	N.	Size (mm)
Group 1	Plain Concrete (PC)	3	150 × 300	4	150 × 300
Group 2	Waste Ceramic Optimal Concrete (WOC)	3	150 × 300	4	150 × 300
Group 3	WOC-Hybrid concrete	3	150 × 300	4	150 × 300

**Table 5 materials-15-02371-t005:** The total used symbols in the present study.

Symbols	Nomenclature
25PC	M25 Plain Concrete
25PC-UH	M25 Un-Heated Plain Concrete
25PC-H-100 °C	M25 Heated Plain Concrete Under Temperature 100 °C
25PC-H-200 °C	M25 Heated Plain Concrete Under Temperature 200 °C
25PC-H-300 °C	M25 Heated Plain Concrete Under Temperature 300 °C
25WOC	M25 Waste Ceramic Optimal Concrete
25WOC-UH	M25 Un-Heated Waste Ceramic Optimal Concrete
25WOC-H-100 °C	M25 Heated Waste Ceramic Optimal Concrete Under Temperature 100 °C
25WOC-H-200 °C	M25 Heated Waste Ceramic Optimal Concrete Under Temperature 200 °C
25WOC-H-300 °C	M25 Heated Waste Ceramic Optimal Concrete Under Temperature 300 °C
25WOC-Hybrid-UH	M25 Waste Ceramic Optimal Hybrid Fiber Concrete
25WOC-Hybrid-H-100 °C	M25 Heated Waste Ceramic Optimal Hybrid Fiber Concrete Under Temperature 100 °C
25WOC-Hybrid-H-200 °C	M25 Heated Waste Ceramic Optimal Hybrid Fiber Concrete Under Temperature 200 °C
25WOC-Hybrid-H-300 °C	M25 Heated Waste Ceramic Optimal Hybrid Fiber Concrete Under Temperature 300 °C

**Table 6 materials-15-02371-t006:** Mix proportion.

Mix Ingredients (kg/m^3^)			
Material	PC	WOC	WOC-Hybrid
Water	190	190	190
OPC (43 grad)	380	342	342
C_WC_	-	38	38
NCA	1118	894	894
A_WC_	-	224	224
Sand	609	548	548
S_WC_	-	61	61
Weight Proportion Fiber (by Volume of Concrete)			
PVA-1%	-	-	13
CR-1%	-	-	78

**Table 7 materials-15-02371-t007:** Average compressive and tensile strength for various mixes.

S No.	Name	CS	TS
		MPa	MPa
1	PC-UH (Reference)	27.99	3.69
2	PC-H-100 °C	25.12 (−11.16%)	-
3	PC-H-200 °C	20.91 (−18.19%)	3.00 (−19.21%)
4	PC-H-300 °C	19.17 (−36.01%)	2.38 (35.91%)
5	WOC-UH (Reference)	24.64	2.68
6	WOC-H-100 °C	22.2 (−8.48%)	-
7	WOC-H-200 °C	19.42 (−16.72%)	2.30 (14.18%)
8	WOC-H-300 °C	17.91 (−34.25%)	1.86 (30.06%)
9	WOC-Hybrid-UH (Reference)	28.80	4.97
10	WOC-Hybrid-H-100 °C	27.00 (−7.15%)	-
11	WOC-Hybrid-H-200 °C	25.45 (−14.55%)	4.46 (−10.26%)
12	WOC-Hybrid-H-300 °C	22.5 (−19.51%)	4.12 (−17.10%)

Note: In Parenthesis is Percentage Decrement. −: decrement.

**Table 8 materials-15-02371-t008:** The input and output parameters used in ANN training.

Samples	Input Parameter			Output Parameter		
	Mix Ingredients (kg/m^3^) (by Weight)	Max	Min	Mechanical Properties	Max	Min
PC	W	190	190	CS (MPa)	27.99	3.69
	NCA	1118	0	TS (MPa)	19.17	2.38
	A_WC_	1118	0			
	NFA	609	609			
	S_WC_	0	0			
	OPC	380	380			
	C_WC_	0	0			
WOC	W	190	190	CS (MPa)	24.64	2.68
	NCA	894	894	TS (MPa)	17.91	1.86
	A_WC_	224	224			
	NFA	609	224			
	S_WC_	426	426			
	OPC	342	342			
	C_WC_	38	38			
WOC-Hybrid	W	190	190	CS (MPa)	28.8	4.97
	NCA	894	894	TS (MPa)	22.5	4.12
	A_WC_	224	224			
	NFA	609	224			
	S_WC_	426	426			
	OPC	342	342			
	C_WC_	38	38			
	CR fiber	156	156			
	PVA fiber	13	13			

**Table 9 materials-15-02371-t009:** Experimental data/results and input and output parameters of PC samples.

Samples	CS (EXP)	CS (ANN)	TS (EXP)	TS (ANN)	Remark
25PC-UH	27.99	27.98876713	3.69	3.6975819	T
25PC-H-100 °C	25.12	25.15282777	-	3.5947	T
25PC-H-200 °C	20.91	20.90893545	3	2.9982176	T
25PC-H-300 °C	19.17	19.17091861	2.38	2.3794473	T
25PC-H-400 °C	-	17.0407	-	1.8614	Test
25PC-H-500 °C	-	15.7647	-	1.5165	Test
25PC-H-600 °C	-	14.5447	-	1.2677	Test
25PC-H-700 °C	-	13.4425	-	1.0705	Test
25PC-H-800 °C	-	12.7729	-	0.9312	Test
25PC-H-900 °C	-	12.478	-	0.8392	Test
25PC-H-1000 °C	-	12.3832	-	0.7796	Test
25WOC-UH	24.64	24.64079669	2.68	2.680382937	T
25WOC-H-100 °C	22.2	22.30830325	-	2.5093	T
25WOC-H-200 °C	19.42	19.43749389	2.3	2.300028182	T
25WOC-H-300 °C	17.91	17.91456212	1.86	1.83237504	T
25WOC-H-400 °C	-	17.0886	-	1.1713	Test
25WOC-H-500 °C	-	16.5171	-	1.2486	Test
25WOC-H-600 °C	-	15.9527	-	0.316	Test
25WOC-H-700 °C	-	15.1142	-	0.3342	Test
25WOC-H-800 °C	-	13.7898	-	0.3342	Test
25WOC-H-900 °C	-	12.023	-	0.297	Test
25WOC-H-1000 °C	-	10.1093	-	0.237	Test
25WOC-Hybrid-UH	28.8	28.8493974	4.97	4.934543642	T
25WOC-Hybrid-H-100 °C	27	26.99951761	-	5.069	T
25WOC-Hybrid-H-200 °C	25.45	25.45138277	4.46	4.461363684	T
25WOC-Hybrid-H-300 °C	22.5	22.56074581	4.12	4.119979713	T
25WOC-Hybrid-H-400 °C	-	20.4875	-	3.6564	Test
25WOC-Hybrid-H-500 °C	-	16.8866	-	3.2572	Test
25WOC-Hybrid-H-600 °C	-	14.1502	-	2.943	Test
25WOC-Hybrid-H-700 °C	-	12.4256	-	2.7155	Test
25WOC-Hybrid-H-800 °C	-	11.089	-	1.56	Test
25WOC-Hybrid-H-900 °C	-	9.7936	-	1.4568	Test
25WOC-Hybrid-H-1000 °C	-	8.6486		1.3878	Test

Training (T) and Testing (Test).

**Table 10 materials-15-02371-t010:** Coefficients of MLR Model.

**Coefficients of Compressive Strength MLR Model**											
	Constant	Water	Cement	Ceramic Powder	Coarse Aggregate	Ceramic Aggregate	Fine Aggregate	Ceramic Sand	CR Fiber	PVA fiber	Temperature
** *Cte* **	16.45	0	0	0	0.0102	0	0	0.0004	0.0255	0	−0.0288
**Coefficients of Tensile Strength MLR Model**											
	Constant	Water	Cement	Ceramic Powder	Coarse Aggregate	Ceramic Aggregate	Fine Aggregate	Ceramic Sand	CR Fiber	PVA Fiber	Temperature
** *Cte* **	−0.045	0	0	0	0.003	0	0	0	0.014	0	−0.004

**Table 11 materials-15-02371-t011:** Formulas of Performance Criteria.

Performance Indicator	Formula
Mean Square Error (MSE)	$\frac{1}{N}\overset{N}{\underset{i=1}{\sum }}{\left(X_{A}^{i}-X_{P}^{i}\right)}^{2}$
Root Mean Square Error (RMSE)	$\sqrt{\frac{1}{N}\overset{N}{\underset{i=1}{\sum }}{\left(X_{A}^{i}-X_{P}^{i}\right)}^{2}}$
Mean Absolute Error (MAE)	$\frac{1}{N}\overset{N}{\underset{i=1}{\sum }}|X_{P}^{i}-X_{A}^{i}|$

**Table 12 materials-15-02371-t012:** Performance criteria for training and test phases.

Model	Indicator	Training	Test
Tensile Strength	R^2^	0.9862	0.9983
	MSE	0.1487	0.2208
	RMSE	0.3856	0.4699
	MAE	0.363	0.469
Compressive Strength	R^2^	0.9715	0.9687
	MSE	0.3931	1.8889
	RMSE	0.6270	1.3744
	MAE	0.4739	1.2297

## Data Availability

The data used to support the findings of this study are included within the article.

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
