# Peer review of "Mechanical Properties, Crack Width, and Propagation of Waste Ceramic Concrete Subjected to Elevated Temperatures: A Comprehensive Study"

_materials, 2022, doi:10.3390/ma15072371_

Round 1

Reviewer 1 Report

The reviewer found the article interesting and well written both for the experimental investigation part and for the presentation of the ANN method. The paper can be pubblished in the present form. The reviewer suggests only to modify and uniform the layout of the figures, because they are presented in different format. For example, Fig 1 and 2 are made by excel with a certain style, as well as chart 1 and 2. Fig. 16 is made by matlab with another style. Figs 17 and 18 again with excel with another style and so on. It is therefore suggested to uniform the format and the style of the figures.

Reviewer 2 Report

This research is interesting and well-organized, which investigates the unconfined compressive strength and tensile strength for plain concrete (PC), waste ceramic optimal concrete (WOC), and Hybrid fiber reinforced waste ceramic optimal concrete (Hybrid-WOC). The experimental part of this paper is interesting and meaningful, which is worth being published. However, some issues, especially ML modeling, must be addressed before publication.

1. The abstract is too long, around 394 words. Only most important findings and conclusions or motivations are needed to be written in this part.
2. The introduction part can be improved. The following references are helpful (Fracture behavior of a sustainable material: Recycled concrete with waste crumb rubber subjected to elevated temperatures; A Multi-objective Optimisation Approach for Activity Excitation of Waste Glass Mortar).
3. The architecture of ANN models in this study is one hidden layer, please explain why you choose one hidden layer? Normally, 1-3 hidden layer is compared to seek the optimal architecture. Also neurons in each hidden layer.
4. One big problem is the database. The data in this study shown in table 7 is very limited. It is mentioned that 10 input variables are considered. This is reasonable since more inputs indeed increase the prediction accuracy to a certain degree. However, too many inputs may in turn reduce the accuracy when the database is very small. Please explain if more data are used in this study.
5. Following the previous comment, did the authors try LR or MLR to predict the strength? In this case, I think the ANN seems unnecessary to be used since LR or MLR might give good predictions as well.
6. The conclusion is also too long, as well as the whole paper. I suggest reducing some parts of this paper to make it brief and clear.
